# Proteomics Analysis of Peripheral Blood Mononuclear Cells from Patients in Early Dengue Infection Reveals Potential Markers of Subsequent Fluid Leakage

**DOI:** 10.3390/v17060805

**Published:** 2025-05-31

**Authors:** Nilanka Perera, Abhinav Kumar, Bevin Gangadharan, Diyanath Ranasinghe, Ananda Wijewickrama, Gathsaurie Neelika Malavige, Joanna L. Miller, Nicole Zitzmann

**Affiliations:** 1Department of Biochemistry, University of Oxford, South Parks Road, Oxford OX1 3QU, UK; nilanka@sjp.ac.lk (N.P.); abhinav.bioc1213@gmail.com (A.K.); bevin.gangadharan@bioch.ox.ac.uk (B.G.);; 2Department of Medicine, Faculty of Medical Sciences, University of Sri Jayewardenepura, Nugegoda 10250, Sri Lanka; 3Kavli Institute for Nanoscience Discovery, University of Oxford, South Parks Road, Oxford OX1 3QU, UK; 4Department of Immunology and Molecular Medicine, Faculty of Medical Sciences, University of Sri Jayewardenepura, Nugegoda 10250, Sri Lanka; diyanath91@gmail.com (D.R.); neelika@sjp.ac.lk (G.N.M.); 5National Institute of Infectious Diseases, Angoda 10620, Sri Lanka; anandawijewickrama012@gmail.com

**Keywords:** dengue, proteomics, dengue haemorrhagic fever, fluid leakage, disease severity, markers

## Abstract

Infections caused by dengue virus (DENV) result in significant morbidity and mortality. A proportion of infected individuals develop dengue haemorrhagic fever (DHF) characterized by circulatory collapse and multiorgan failure. Early detection of individuals likely to develop DHF could lead to improved outcomes for patients and help us use healthcare resources more efficiently. We identified proteins that are differentially regulated during early disease in peripheral blood mononuclear cells (PBMCs) of patients who subsequently developed DHF. Four dengue fever (DF), four DHF and two healthy control PBMCs were subjected to tandem mass tag mass spectrometry. Differentially regulated proteins were used to identify up- or down-regulated Gene Ontology pathways. One hundred and sixty proteins were differentially expressed in DENV-infected samples compared to healthy controls. PBMCs from DHF patients differentially expressed 90 proteins compared to DF; these were involved in down-regulation of platelet activation and aggregation, cell adhesion, and cytoskeleton arrangement pathways. Proteins involved in oxidative stress and p38 MAPK signalling were upregulated in DHF samples during early infection compared to DF. This study has identified 90 proteins differentially regulated in PBMCs that could potentially serve as biomarkers to identify patients at risk of developing DHF at an early disease stage.

## 1. Introduction

Dengue virus (DENV) infections are prevalent throughout the world with an estimated 390 million cases occurring every year [1]. Asia harbors approximately seventy percent of this viral infection which causes significant morbidity and mortality [1]. DENV infections account for 1.1 million disability-adjusted life-years globally [2]. In the majority of cases, dengue fever (DF) is characterized by an initial febrile phase with viraemia, followed by resolution of fever leading to recovery [3]. A minority of DENV-infected individuals develop plasma leakage at the time of defervescence, known as the “critical phase” which typically lasts 24–48 h [4]. The increased vascular permeability causing plasma leakage is reflected by the development of serosal effusions and a rise in haematocrit [5]. Such dengue haemorrhagic fever (DHF) patients, unless resuscitated with adequate fluids, develop shock, leading to multiorgan failure and coagulopathy, which causes significant mortality [4,6]. Patients with severe plasma leakage (leading to shock or respiratory distress), severe bleeding and severe organ impairment are considered as severe dengue patients [7]. Prevalence of DHF varies from 6.3% in DENV-infected individuals (irrespective of symptoms) to 45.7% in symptomatic and hospitalized cases [8]. There are no clinical or biochemical markers at present that accurately predict the development of DHF during early infection. Thus, stringent monitoring and regular bedside ultrasonography in healthcare facilities are required to identify patients developing DHF. Inability to triage patients during early infection results in overcrowding of healthcare facilities and increased utilization of resources.

The pathogenesis of plasma leakage is presumed to be due to alterations in the endothelial glycocalyx. Studies suggest a possible role of DENV non-structural protein 1 (NS1) and its interactions with the endothelium [9,10], although exactly how this happens is unknown. The roles of many cytokines and vascular adhesion markers have been studied in severe disease, though with limited utility in clinical practice [11]. In addition, increased endothelial cell permeability could be due to complex interactions between DENV-infected cells such as monocytes, platelets and the vascular endothelium. Host immune responses in dengue infection, consisting of innate immune responses and adaptive immunity, play an important role in disease pathogenesis [12,13]. Monocytes, tissue macrophages and dendritic cells are key players of the innate immune response, which recognize the virus by pattern-recognition receptors, leading to production of downstream inflammatory mediators [12,14,15]. Therefore, dengue-infected mononuclear cells provide an important platform to identify disease severity markers. Some studies have explored the transcriptome of DENV-infected peripheral blood mononuclear cells (PBMCs) in an effort to identify differential gene signatures leading to severe disease [16,17,18]. Although such analysis may provide useful insights, disease usually manifests on the protein rather than gene level, and identification of key proteins is also more suitable for developing tests for relevant markers. Studies have explored serum or plasma proteomes of dengue-infected patients in an attempt to identify markers indicating severe disease [19,20,21,22]. Plasma proteomics could be challenging when identifying low-abundance proteins [23]. Cellular proteomics of infected immune cells provide a useful platform to explore markers of dengue disease severity. Although immune responses have been profiled previously, the proteome of PBMC lysates of dengue-infected patients of different disease severity is yet unexplored. The proteome of infected mononuclear cells in patients with DHF during early infection could also provide valuable information on the pathways triggered at that point that may subsequently lead to plasma leakage. Here, we report the results of a tandem mass tag (TMT) mass spectrometry experiment and comprehensive analysis aimed at identifying differentially regulated proteins in patients with DF compared to those who subsequently went on to develop DHF, using samples of healthy individuals as controls. The present study identifies previously under-recognized biological pathways and individual proteins potentially involved in causing plasma leakage, laying the foundation for future validation studies.

## 2. Materials and Methods

### 2.1. Study Setting and Patient Recruitment

A total of 25 dengue patients fulfilled the inclusion criteria and they were recruited from the National Institute of Infectious Diseases, Sri Lanka. Of these patients, 17 had DF and 8 patients progressed to DHF in the subsequent days. The study participants were recruited from the outpatient department and the wards. The criterion for inclusion in the study was patients with a febrile illness suggestive of dengue fever lasting less than 72 h. Patients less than 18 years of age were excluded due to complexities involved in obtaining consent. In addition, pregnant patients were also excluded due to diverse factors affecting the course of illness. Any patient with evidence of fluid leakage on recruitment was excluded from the study to facilitate recruitment of patients who were at an early disease stage. The SD Bioline dengue rapid NS1 kit (Abbott, Chicago, IL, USA) was used to identify patients with dengue fever on recruitment.

The study was explained to the participants and written consent was obtained at recruitment. After obtaining their consent, one venous blood sample from each participant was taken at admission, representing the initial febrile phase of dengue infection. A 5 mL blood sample was collected into a 15 mL heparinised container for PBMC isolation and 2 mL of blood was collected in a plain tube for separating serum by centrifugation at 2000 rpm. All participants had the right to withdraw from the research during the entire study period. Clinical data were obtained from the bed-head tickets and samples were initially processed in the Centre for Dengue Research at the University of Sri Jayewardenepura, and final proteomics experiments were carried out at the Oxford Glycobiology Institute, University of Oxford. Fifteen healthy volunteers were recruited to generate samples to be used as controls from Sri Lanka during the study period.

### 2.2. Measurement of Viral Load in Serum

RNA extraction of serum samples was performed using the QiaAmp viral RNA mini kit (Qiagen, Hilden Germany) using 140 μL of serum according to the manufacturer’s instructions. A detailed description is provided in Appendix A.

### 2.3. Measurement of Dengue-Specific IgG in Serum

A Panbio^TM^ dengue IgG indirect ELISA kit (01PE30, Abbott, Illinois, USA) was used to detect dengue-specific IgG levels in the serum according to the manufacturer’s instructions.

### 2.4. PBMC Isolation

The PBMCs were isolated by a gradient centrifugation method. A detailed description is provided in Appendix A.

### 2.5. Tandem Mass Tagging

PBMC cell lysates were produced by resuspending the cell pellets in RIPA buffer, supplemented with protease inhibitor (Roche, Basel, Switzerland) and phosphatase inhibitor (Sigma, Saint Louis, MO, USA), and incubating them at 4 °C for 20 min. A 10-plex tandem mass tagging was performed. A detailed description is provided in Appendix A.

### 2.6. Liquid Chromatography

The 11 TMT labelled peptide fractions were separated on a Dionex Ultimate 3000 nano UHPLC system (ThermoFisher Scientific, Waltham, MA, USA). A detailed description is provided in Appendix A.

### 2.7. Mass Spectrometry

Peptides from the nano LC were analysed on a benchtop Q Exactive hybrid quadrupole–Orbitrap mass spectrometer using the Nanospray Flex ion source (ThermoFisher Scientific). A detailed description is provided in Appendix A.

### 2.8. Protein Identification

The acquired *.raw file for the 5 μL labelling check was converted to a Mascot generic file (*.mgf) using MSConvertGUI 64-bit (ProteoWizard). A peak-picking filter was set with MS levels 1–2. The Mascot server software (Matrix Science, London, UK) was used to search the *.mgf file against the SwissProt database. A detailed description is provided in Appendix A.

### 2.9. Proteomic Data Analysis

All human proteins identified in the samples were selected for data analysis. The average value of the MS2 tag peak intensity was calculated for proteins with multiple peptides. The mean MS2 tag peak intensity ratio for each peptide was calculated for the healthy, DF and DHF samples. The H1 healthy sample was used as the reference sample to calculate this ratio as the relative MS2 tag peak intensity for each sample. The fold change for each protein was calculated for dengue-infected samples (DF and DHF both) compared to healthy controls (H1 and H2) and DF (DF 1–4) compared to DHF (DHF 1–4) using average values for each group. A *t*-test was performed among samples to identify a statistically significant difference in protein expression between healthy controls and dengue-infected samples. A similar analysis was performed for DF and DHF. Proteins with a fold difference exceeding 1.5 and *p* < 0.05 were selected to identify biological pathways associated with the proteins using the STRING version 12.0 database. Figures were generated using GraphPad Prism version 9.0 and R studio with R v4.1.2. Protein to protein interaction (PPI) networks were created using Cytoscape v3.9.1. Gene ontology enrichment analysis was conducted by calculating probabilities and fold enrichment values from observed differently expressed proteins. The Gene Ontology (GO) database [24] was used to enrich biological processes, molecular function and cellular components, while the Kyoto Encyclopedia of Genes and Genomes (KEGG) and REACTOME databases were used to enrich pathways [25,26]. Fold enrichment and false discovery rate (FDR) were calculated for each pathway. Fold enrichment measures how drastically the genes of a certain pathway is overrepresented, which was calculated by the percentage of differently expressed proteins belonging to a pathway divided by the corresponding percentage in the background. FDR was calculated based on the nominal *p*-value from the hypergeometric test, which represents the statistical probability of each enrichment. A cut-off of 0.05 was used on the FDR to filter enriched pathways. Calculations were performed using R\GO.db and R\Gostats packages, and visualizations were performed using R\ ggplot2.

## 3. Results

We recruited 25 patients (17 DF and 8 patients who subsequently developed DHF) to the study who fulfilled the selection criteria early during the course of the disease and collected their PBMCs to be lysed. The protein content of the lysates was quantified and the four DF and four DHF samples with the highest protein content were selected for subsequent TMT mass spectrometry experiments. This approach was used to enhance protein identification from the TMT experiment and selection was not based on clinical characteristics to avoid bias. Selected samples in each group (DF 4/17 and DHF 4/8) were representative of the total study participants recruited to the study with regard to demography and outcome.

PBMC lysates from two healthy controls and four DF and four DHF patients were analysed by TMT mass spectrometry. The sera of the selected patients were positive for DENV viral genome identified by qRT-PCR. All patients (n = 8) were infected by DENV serotypes 2 and 3 DF (75%), and all DHF patients had detectable IgG levels in the initial serum sample suggestive of secondary dengue infection. Patients who developed DHF had evidence of fluid leakage (pleural effusion and/or ascites) during their later course of illness. The characteristics of the patients are given in Table 1. DHF patients had a significantly lower platelet count over the course of illness. Other parameters were not significantly different between the two groups. Healthy controls (n = 2) included a male/female ratio of 1:1 and mean age of 27.5 years (SD 5.0).

The initial check for successful TMT labelling of the unfractionated PBMC lysate digest revealed 1018 proteins labelled with TMT and 15 proteins that were not labelled, confirming that 98.5% of the peptides identified were successfully labelled. The mass spectrometry analysis of the fractionated digest identified 7352 human peptides (FDR < 1%) corresponding to 1931 proteins. DENV proteins were not identified in any of the samples. We compared the proteome of two healthy controls, four DF patients (without evidence of plasma leakage) and four DHF patients (with plasma leakage). One healthy control (H1) was considered the reference sample, and fold changes in protein expression in all the samples were calculated based on this reference sample. The mean fold change was calculated for the groups (healthy, DF and DHF) and a *t*-test was used to find a significant association among groups. Proteins with a significant fold change (>1.5) and *p* < 0.05 were considered differentially expressed during group comparisons.

### 3.1. Proteins Differentially Regulated in Dengue-Infected PBMCs Compared to Healthy Controls

We identified 160 differentially regulated proteins (red category in Figure 1a), including 94 upregulated and 66 down-regulated proteins, with a fold change of >1.5 and *p* < 0.05, when comparing the eight DENV-infected samples with healthy control samples. There were 210 proteins which had a fold change >1.5, which were, however, not statistically significant, and 502 proteins that had a statistically significant fold change <1.5 (green and blue categories, respectively, in Figure 1a). These proteins were further analysed to identify the main gene ontology (GO) pathways involved in DENV-infected PBMC,, to determine the number of proteins involved and the fold enrichment (Figure 2). GO enrichment analysis revealed 197 (FDR ≤ 0.05) pathways upregulated and 122 (FDR ≤ 0.05) pathways down-regulated in DENV-infected PBMCs. Response to cytokines, cytokine-mediated signalling pathways, type 1 interferon signalling and defence response to virus were upregulated as expected in DENV-infected PBMCs during early infection. Most upregulated proteins were predicted to be localized in the MHC class I peptide loading complex and postsynaptic endocytic zone in the cells. All proteins differentially regulated in DENV-infected samples are listed in Appendix A. The heatmap and the pathway analysis are shown in Figure 3.

### 3.2. Proteins Differentially Regulated in DHF Compared to DF

Ninety proteins were differentially expressed in DHF compared to DF PBMCs with a fold change of >1.5 and *p* < 0.05 (red category in Figure 1b). There were 278 proteins which had a fold change >1.5, which were, however, not statistically significant, and 232 proteins that had a statistically significant fold change <1.5 (green and blue categories, respectively, in Figure 1b). In total, 12 proteins had increased and 78 had decreased expression levels in DHF lysates (Figure 4). GO enrichment revealed a total of 142 upregulated and 150 down-regulated pathways. Highly enriched pathways in DHF were involved in negative regulation of macrophage activation and interleukin-8 production, as well as positive regulation of the p38 MAP kinase cascade and ERBB signalling. Most upregulated proteins were localized in phagolysosomes, lysosomes and the NADPH oxidase complex (Figure 5). Down-regulated proteins were mainly involved in coagulation which included platelet activation and aggregation. Many down-regulated proteins were localized at cell–substrate junctions, focal adhesion and anchoring junctions in the cells. All differentially expressed proteins are listed in Appendix A. Selected proteins of interest are shown in Appendix A.

## 4. Discussion

DHF can be fatal if not managed appropriately. Due to a lack of markers that could accurately predict development of plasma leakage, patients with significant thrombocytopaenia are admitted to hospitals and monitored closely. This results in significant overcrowding of hospitals in many endemic countries with overburdening of staff. Therefore, identification of biomarkers that could accurately predict disease severity in dengue fever patients early during the course of disease has been at the heart of much ongoing research. Variable and non-reproducible results have been a problem in utilizing, for example, inflammatory mediators as potential markers. Previous transcriptomic analysis of patient PBMCs identified increased expression of RNA transcripts involved in mitochondrial processes and neutrophil-associated enzymes (MO, ELANE) in severe dengue [16,17]. Nikolayeva et al. identified an 18-gene panel in PBMCs for detection of severe dengue on admission to hospital [27]. A study by Jose et al. revealed that reduced expression of indoleamine 2,3-dioxygenase (IDO) pathway genes such as IDO1, AhR and TGFβ were significantly associated with severe disease [28]. MicroRNA expression levels have also been explored in the peripheral blood cells to predict severe dengue infection in previous studies in addition to many genetic panels [29,30]. Although transcriptomic data provide useful information, identification of proteins provides a more robust indication of cellular activity and relevant proteins identified may also be useful in developing cost-effective lateral flow tests. Platelet activation and sequestration of histones were seen in platelets of dengue-infected samples in a proteomic analysis [31]. This proteomic analysis revealed important observations on the role of platelets in dengue infection, but it was not designed to find an association with disease severity. Currently, there are no approved biomarkers that could be used to triage patients in early disease. Here, we report for the first time which proteins are differentially expressed in PBMCs of DHF patients compared to those with non-severe disease. This study was performed using samples of patients during the early disease stage, before they developed plasma leakage. The proteins identified here could therefore, once validated in a larger cohort, potentially be useful biomarkers for detecting DHF early in the disease course, in addition to shedding light on pathways potentially contributing to plasma leakage.

We did not identify any DENV proteins in the PBMC samples. This is not surprising due to the low levels of virus proteins expected in the infected cells. Dengue-infected PBMCs showed increased expression levels of proteins involved in interferon signalling compared to healthy control samples as observed previously in transcriptomic analysis of DENV-infected PBMCs [16,17]. Type I interferons elicit antiviral responses to protect the host through many interferon-stimulated genes (ISGs) and the ISG15 protein plays a central role in this defence response [32]. ISG15 was highly expressed in DENV-infected PBMCs compared to healthy controls indicating the type I interferon-driven antiviral response. However, ISG15 was not differently expressed in DHF compared to DF which confirms similar findings from previously published transcriptomic data [17]. In addition, proteins involved in inflammatory signalling (which includes cytokine activation) and proteins localized in the phagocytic vesicles were also expressed at higher levels in DENV-infected PBMCs to combat the virus. Proteins involved in the cytoskeleton, cell–substrate junction and actin-filament-based processes were down-regulated in dengue-infected PBMCs. This reduced expression of proteins of the cytoskeleton and cell adhesion was more pronounced in patients who developed DHF compared to DF. The host cytoskeleton is known to play an important role in facilitating DENV entry to cells and viral replication [33]. DHF PBMCs exhibited down-regulation of these processes, suggesting a yet unidentified role in the development of vascular leakage, possibly by leucocyte–endothelial cell interaction. Endothelial cells infected in vitro with DENV and treated with TNFα showed reduced levels of moesin (a cytoskeletal protein), resulting in increased trans-endothelial permeability, as reported previously [34]. However, the exact mechanisms and cascade of events leading to increased vascular permeability are still unknown.

A negative effect on proteins involved in haemostasis, platelet activation, aggregation and signalling was observed in dengue-infected PBMCs, and this down-regulation was more pronounced in the DHF samples compared to DF lysates. Thrombocytopaenia is known to be more pronounced in DHF and it predicts a worse outcome [35]. Many proteins such as TLN1 [36], FLNA [37], and VWF [38], important in haemostasis and platelet function, were down-regulated in DHF PBMCs during the early disease course, suggesting that bleeding manifestations seen in DHF could partly be due to these suppressed pathways. In addition, FXIIIA (factor 13 subunit A), which circulates in plasma, was significantly reduced in PBMCs of patients who progressed to DHF. FXIII has two subunits, of which the A subunit is secreted from cells of myeloid lineage and platelets [39]. Subunit B is secreted from the hepatocytes [39]. Although the main function of FXIII is to stabilise the clot, many other functions of this coagulation factor in monocytes are recognized as being mainly related to phagocytosis and interaction with the cytoskeleton [40]. Plasma levels of FXIIIA can be measured and it could be a useful marker to identify development of DHF in future studies.

Two important observations were the increased activation of oxidative stress-related pathways and activation of p38 mitogen-activated protein (MAP) kinase signalling in DHF samples. MAP kinases are important intracellular kinases involved in regulation of cellular proliferation, differentiation, and apoptosis in addition to their role in cellular inflammation [41]. p38 MAP kinase activation results in production of pro-inflammatory cytokines such as IL-1β, IL-6, and TNFα [42], and p38 has been reported to regulate TNFα gene transcription [43]. In vitro studies have shown that DENV-infected endothelial cells express increased amounts of p38 MAP kinases, and it is postulated that p38 signalling increases levels of pro-inflammatory mediators such as IL-8 [44]. p38 MAP kinase was also activated in endothelial cells by DENV NS1 protein, resulting in increased endothelial permeability [9] and inhibition of p38 signalling in vivo, reduced inflammatory response, vascular leak and improved survival of mice [45]. Cross-talk between dengue-infected immune cells and the endothelium could be important in the disruption of endothelial integrity and the development of vascular leak. Thus, findings on the PBMCs of patients who subsequently developed DHF indicate a p38 MAP kinase-driven response potentially contributing to severe disease. In addition, proteins involved in this pathway could be developed into early biomarkers predicting severe disease.

Oxidative stress is known to play a role in DENV pathogenesis [46]. Reactive oxygen species (ROS) activate inflammatory pathways in immune cells [46] and induce vascular permeability in endothelial cells [47]. Markers of oxidative stress such as malondialdehyde are elevated in DHF [48]. Previous studies have also demonstrated the increased expression of inducible nitric oxide synthase (iNOS) genes in serum of patients developing DHF [49]. Upregulated proteins in DHF samples implicated in oxidative stress could be useful as markers of fluid leakage following validation in larger cohorts.

Neutrophil-associated transcripts were abundant in previous transcriptomic analysis of PBMCs of patients who developed dengue shock syndrome [18]. Consistent with this study, we identified BPI (bactericidal permeability increasing) protein to be high in the DHF patient PBMCs. The role of neutrophils in dengue infection is not clear. Studies have shown that dengue infection results in in vivo neutrophil activation and neutrophil extracellular trap formation [50,51]. The authors hypothesized that a possible role of neutrophil extracellular traps may be in mediating vascular leakage by disrupting the endothelial barrier [50]. NGAL (Neutrophil Gelatinase-associated Lipocalin) is a protein initially purified from neutrophils but expressed in many organs and cells. A previous study showed a higher serum level of NGAL in dengue-infected individuals compared to healthy controls [52]. Levels correlated with white blood cells and platelets. However, a relationship to disease severity was not investigated in this study. NGAL is one of the few proteins highly expressed in PBMCs of patients who progress to DHF compared to DF. Therefore, these proteins could be potential markers to identify DHF early in the disease course in future validation studies.

Although not performing validation of the identified markers is a limitation of our study, we have provided a platform to design future studies for identifying DHF early in the disease course. Validation of the identified markers could be performed in the serum of patients using an antibody-based method, such as ELISA, Western blotting or immunoprecipitation–mass spectrometry (IP-MS), which could later lead to the development of a lateral flow test for any promising marker in the clinical setting. An adequately powered study recruiting patients in the early phase of infection would pave the way for finding a potentially relevant biomarker capable of identifying patients likely to develop fluid leakage. Many of the identified markers and pathways would shed light on disease pathogenesis in addition to serving as potential diagnostics. We were limited to using 10 samples to discover the biomarkers since we used a TMT 10-plex kit. Although this sample size is small, this number of samples is commonly used for TMT-based biomarker discovery and allows for the simultaneous quantification of proteins across all 10 samples. More samples could have been run using a label-free approach for biomarker discovery but TMT affords higher precision, better relative quantitation, more reliable normalization, and better comparability across samples and overcomes inter-run variability and instrument drift, which is a problem, especially since each sample was fractionated into 11 fractions using high-pH reversed-phase fractionation (Appendix A). On average, the DHF group was older than the DF group, although this difference was not statistically significant. The difference in age could potentially affect the proteome of the group analysed. Future validation studies would benefit from matching the demographic parameters to avoid such potential bias.

In conclusion, we have for the first time explored the proteome of PBMCs from DENV-infected patients during early infection to identify potential markers predicting DHF as well as pathways implicated in vascular leakage. Our study was aimed at identifying proteins from PBMC lysates to gain a better understanding how processes within these cells may cause, reflect and perhaps also predict the further course of disease. To develop promising proteins into useful biomarkers, a next step will be to look at the secretion of selected proteins into the bloodstream and validate them in larger cohorts as potential biomarkers to predict the risk of developing DHF during early stages of dengue infection.

## Figures and Tables

**Figure 1 viruses-17-00805-f001:**
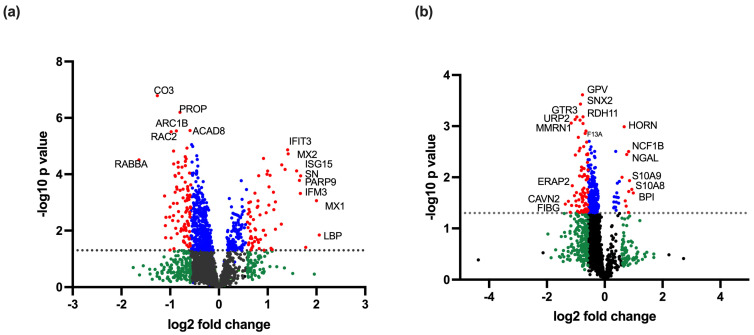
Volcano plots of (**a**) eight dengue-infected samples compared to two healthy controls and (**b**) four DHF compared to four DF samples. In all cases the mean values of each protein expression were compared. The plots show the fold change (log_2_ fold change) and the *p* value following a *t*-test (-log_10_ *p* value) of the expression level of each protein detected in PBMC lysates as analysed by TMT mass spectrometry. Black represents proteins with no difference in expression; green represents a fold change difference of >1.5 without a statistically significant *p* value (*p* > 0.05); blue represents a fold change of <1.5 with a statistically significant *p* value (*p* < 0.05); red represents proteins with a fold change of >1.5 and a statistically significant difference (*p* < 0.05) among groups. Proteins having high -log_10_ *p* value and log_2_ fold change are labelled with their names.

**Figure 2 viruses-17-00805-f002:**
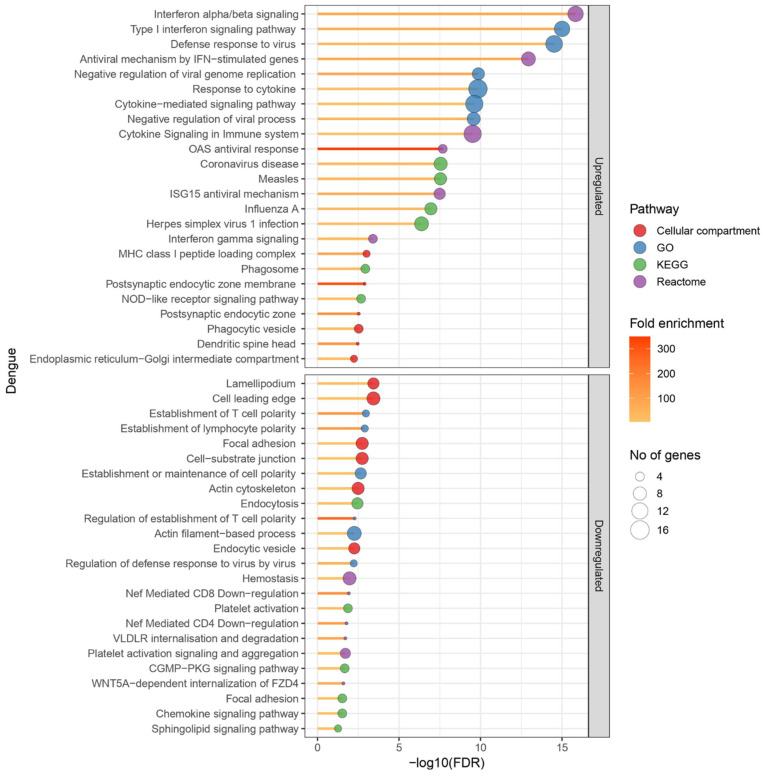
Gene ontology (GO) pathways identified to be differentially regulated in the dengue-infected PBMC lysates compared to healthy controls. Two healthy control PBMC lysates and eight dengue-infected lysates were subjected to TMT mass spectrometry. Mean protein expression levels among healthy samples were compared to those of dengue-infected samples. Proteins with a fold change of >1.5 and a statistically significant difference in dengue-infected lysates compared to healthy controls were subjected to STRING pathway analysis. Main pathways differentially regulated in dengue-infected lysates are given in the figure. The size of the blue circle corresponds to the number of proteins involved in each pathway and the yellow-red colour code corresponds to the fold enrichment. GraphPad Prism version 9.0 and R studio with R v4.1.2 were used to create the figure.

**Figure 3 viruses-17-00805-f003:**
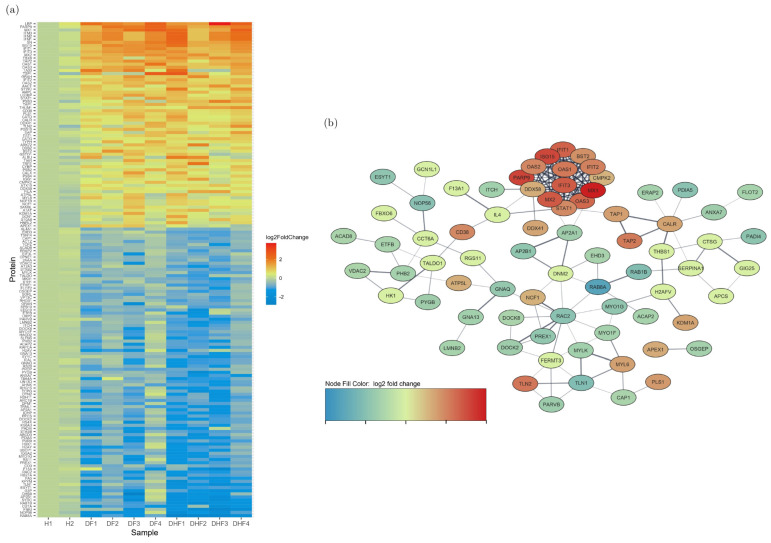
Differentially regulated proteins in dengue-infected PBMC lysates compared to the healthy controls. (**a**) The 160 proteins expressed to a significantly lower or higher level in the DV-infected PBMC lysates (DF 1–4 and DHF 1–4) compared to the healthy control sample 1 (H1) are shown in the heat map. The red colour indicates a higher expression level while the blue colour indicates a lower expression level in dengue samples compared to H1. (**b**) The protein network of the differentially regulated proteins identified in the DV-infected samples by STRING software.

**Figure 4 viruses-17-00805-f004:**
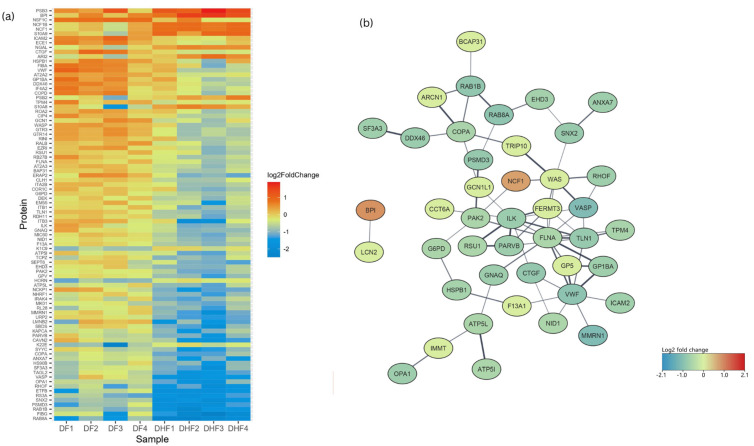
Differentially regulated proteins in dengue fever (DF) patient PBMC lysates compared to dengue haemorrhagic fever (DHF). (**a**) The 90 proteins that were expressed at significantly lower or higher levels in DF compared to DHF PBMC lysates (DF 1–4 vs. DHF 1–4) after normalization to healthy control sample 1 (H1) are shown in a heat map. The red colour indicates a higher expression level while the blue colour indicates a lower expression level among DHF samples compared to DF sample 1. (**b**) Protein network of the differentially regulated proteins identified in the DHF samples compared to DF samples as analysed by STRING software.

**Figure 5 viruses-17-00805-f005:**
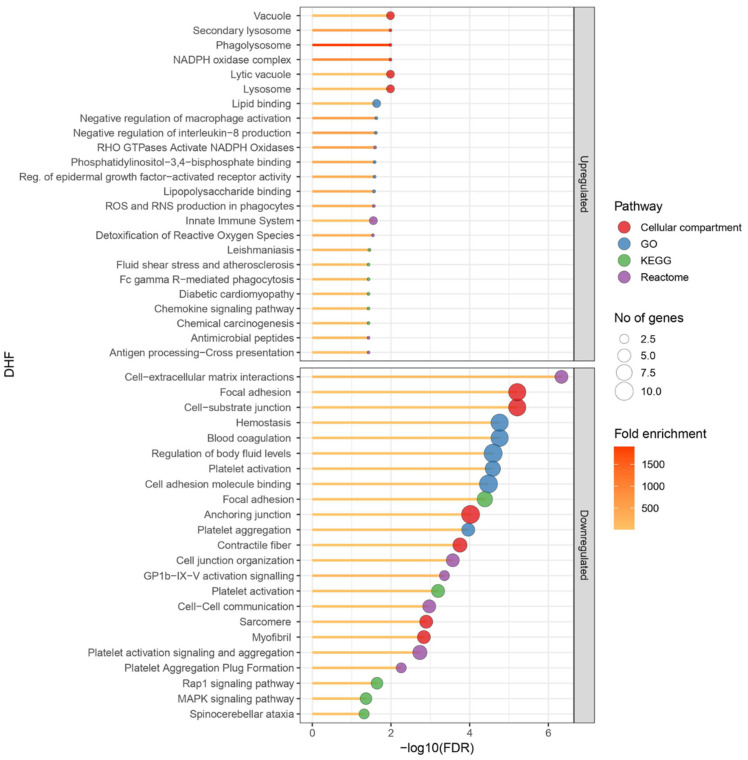
Gene ontology (GO) pathways identified to be differentially regulated in the dengue haemorrhagic fever (DHF) patient PBMC lysates compared to dengue fever (DF) samples. Four DF patient PBMC lysates and four DHF patient lysates were subjected to TMT mass spectrometry. Mean protein expression levels of DF samples were compared to those of the DHF samples. Proteins with a fold change of >1.5 and a statistically significant difference in DHF patient lysates compared to DF samples were subjected to STRING pathway analysis. The main differentially regulated pathways are shown in the figure. The size of the blue circle corresponds to the number of proteins involved in each pathway and the yellow-red colour code corresponds to the fold enrichment. GraphPad Prism version 9.0 and R studio with R v4.1.2 were used to create the figure.

**Table 1 viruses-17-00805-t001:** Characteristics of the patients selected for the TMT mass spectrometry study.

Clinical Parameter	DF n = 4	DHF n = 4	Significance (*p*-Value)
Age in years, mean (SD)	25.7 (7.9)	38.8 (8.8)	0.07
Males (n, %)	2 (50)	2 (50)	1.0
Laboratory investigations on admission in the febrile phase, mean (SD)			
White cells (×10^9^/L)	6.2 (3.3)	5.5 (1.46)	0.71
Platelets (×10^9^/L)	189 (82)	193 (62)	0.94
Packed cell volume (%)	39.9 (3.2)	39.2 (5.1)	0.82
Highest/lowest values of laboratory investigations during the course of illness, mean (SD)			
Lowest white cells (×10^9^/L)	2.7 (0.4)	4.4 (1.5)	0.07
Lowest platelets (×10^9^/L) *	70.5 (37.2–157)	13.5 (12.2–26)	0.03
Highest packed cell volume (%)	42.2 (5.2)	45.2 (7.5)	0.53
Liver enzymes, median (IQR)			
AST (U/L)	68.7 (30.4–157)	65 (51.9–394.2)	0.56
ALT (U/L)	65.1 (40.6–154.7)	54.8 (28.9–339.5)	0.89
DENV diagnosis, n(%)			
Positive NS1 rapid kit	4 (100)	4 (100)	1.0
Virus detected by qRT-PCR	4 (100)	4 (100)	1.0
Viral load in the acute phase (log_10_ GE) *	2.1 (1.5–5.0)	1.9 (1.4–2.3)	0.38

* Values given as median (IQR).

## Data Availability

The datasets used and analysed in the current study are available from the corresponding author on reasonable request.

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
