# Peer review of "Proteomics Analysis of Peripheral Blood Mononuclear Cells from Patients in Early Dengue Infection Reveals Potential Markers of Subsequent Fluid Leakage"

_viruses, 2025, doi:10.3390/v17060805_

Round 1
Reviewer 1 Report
Comments and Suggestions for Authors
This study investigates early molecular markers that could predict which dengue patients are at risk of progressing to dengue haemorrhagic fever (DHF), a severe and potentially fatal complication characterized by plasma leakage. Using proteomic analysis of peripheral blood mononuclear cells (PBMCs) from patients in the early phase of dengue infection, the authors identify differentially expressed proteins and pathways that may serve as early biomarkers for DHF.
The work addresses a critical gap in dengue management for early identification of patients at risk for severe disease. By focusing on the PBMC proteome rather than serum or transcriptome alone, the study targets the cellular level where immune responses and disease mechanisms are directly manifested. This approach may yield more actionable biomarkers for clinical use.
Minor concerns:
1. Sample Size: The proteomic analysis was conducted on a small subset (4 DF, 4 DHF, and 2 healthy controls) selected from a larger cohort, which may limit the generalizability and statistical power of the findings.
2. Validation Needed: The study is exploratory and hypothesis-generating. The identified proteins and pathways require validation in larger, independent cohorts before they can be considered for clinical application.
Reviewer 2 Report
Comments and Suggestions for Authors
This study is aiming to investigate the differences in the proteome of PBMCs in dengue infected patients in order to attempt to identify potential markers for severe Dengue disease. Whilst this study is limited to only one dataset from one proteomic analysis it provides a useful starting point in the identification of severe Dengue markers. Whilst the data is interesting further studies are required, and the authors themselves acknowledge that further studies are required to provide any further useful conclusions.
The article is well written and clear with all the useful information provided in the methods. No grammatical errors were detected in the text.
Whilst as mentioned the data is preliminary, the study is providing a useful insight for future work looking into severe dengue disease.
Reviewer 3 Report
Comments and Suggestions for Authors
The manuscript entitled “Proteomics analysis of peripheral blood mononuclear cells from patients in early dengue infection reveals potential markers of subsequent fluid leakage” addresses an important clinical question—the early identification of patients at risk of developing dengue haemorrhagic fever (DHF). By applying tandem mass tag (TMT)-based proteomic profiling to peripheral blood mononuclear cells (PBMCs), the authors aim to identify early molecular markers associated with disease severity. The topic is timely, and the methodological approach is suitable for exploratory biomarker discovery. While the manuscript has several strengths, there are specific areas where clarification and revision are necessary to enhance its scientific validity, clarity, and overall impact.
Major Comments
-
Sample Selection and Size
The proteomic analysis is based on four samples each from DF and DHF patients, selected on the basis of protein yield. I find this selection criterion to be potentially biased and insufficiently justified. The authors should elaborate on the rationale for this approach and clarify whether the selected samples are representative of the larger cohort. -
Description of Control Group
The manuscript does not provide basic demographic or clinical information about the healthy controls. I recommend including details such as age, sex, dengue serostatus (e.g., NS1, IgG), and relevant laboratory parameters. This information is essential for interpreting differences and should be added to Table 1 for clarity. -
Potential Confounding by Age
The DHF group is older (mean age: 38.8 years) than the DF group (mean age: 25.7 years). This age disparity could influence the proteomic profile and should be discussed as a potential confounding factor. -
Statistical Analysis and Multiple Testing
While t-tests were employed to assess differential expression, I did not find any mention of multiple testing correction, such as false discovery rate (FDR) adjustment. This is critical in proteomics studies given the high number of comparisons. The authors should clarify whether FDR or another method was applied, and if not, provide justification. -
Lack of Validation of Proteomic Findings
Although the study identifies candidate biomarkers, no validation was performed using orthogonal techniques (e.g., ELISA, Western blot). While I acknowledge this limitation is mentioned, it should be addressed more explicitly, with an outline of future plans for validation studies.
Minor Comments
-
Language and Clarity
Despite improvements, the manuscript still contains grammatical errors, typographical artifacts (e.g., “weredid not identifiedy”), and verbose constructions. I recommend further professional language editing to improve clarity and readability. -
Spelling and Style Consistency
Inconsistencies persist in the use of British and American English (e.g., “signalling” vs. “signaling”, “analyse” vs. “analyze”). The authors should select one convention and apply it uniformly. In-text references to figures and tables should also be standardized. - Outdated References
A substantial number of citations are over a decade old. While foundational studies are acceptable, I recommend incorporating more recent literature (within the last 5 years), particularly in the background and discussion sections. -
Figure and Table Presentation
Figure legends should be revised to allow for independent interpretation. Table 1 should include statistical comparisons (e.g., p-values) for DF vs. DHF groups, and baseline values for the healthy controls should be added to enable contextual interpretation.
The manuscript is generally comprehensible and presents the scientific findings in a coherent manner. However, there are numerous issues with grammar, punctuation, phrasing, and consistency that affect overall readability and clarity. A careful language review is strongly recommended to improve the quality of expression. Below is a list of the most frequent and notable language and style problems observed:
Common Language, Style, and Grammar Issues
-
Inconsistent Spelling
-
Mixed use of British and American English (e.g., “signalling” vs. “signaling”, “analyse” vs. “analyze”).
➤ A single English convention should be applied throughout the manuscript.
-
-
Grammar and Syntax Errors
-
-
“the exactly mechanism” → should be “the exact mechanism”
-
-
Sentence fragments and improper word order:
-
“due to an alterations…” → should be “due to alterations…”
-
-
-
Redundancy and Wordiness
-
Repetitive or overly verbose constructions:
-
“down-regulation of cytoskeletal proteins of the cytoskeleton”
-
“the identification of proteins provides a more robust indication of cellular activity and proteins are also useful…” – Consider streamlining.
-
-
-
Punctuation and Formatting
-
Multiple spacing, misplaced commas, and inconsistent figure references:
-
“Figure 1a” vs. “figure 1A”
-
Extra spaces before punctuation and inconsistent use of semicolons.
-
-
-
Misuse or Omission of Prepositions and Articles
-
“duringon recruitment” → should be “during recruitment”
-
“compared than into DF” → should be “compared to DF”
-
-
Awkward or Ambiguous Phrasing
-
“This study has identified 90 proteins differentially regulated proteins…” – repeated noun phrase, needs correction.
-
“many other functions are recognized mainly related to phagocytosis…” – unclear subject and verb agreement.
Figure and Table References
-
Inconsistent terminology: “Figure” vs. “figure”; “Supplementary figure 1” vs. “Supplementary Figure S1”.
Consistency and correct capitalization are necessary.
-
-
I recommend that the manuscript undergo thorough professional language editing to resolve the above issues. Doing so will significantly enhance its clarity, precision, and scholarly tone, aligning it with the expectations of an international scientific audience.
